# Prognostic Implications of Insulin Resistance in Heart Failure in Japan

**DOI:** 10.3390/nu16121888

**Published:** 2024-06-14

**Authors:** Keiichiro Iwasaki, Kazufumi Nakamura, Satoshi Akagi, Yoichi Takaya, Hironobu Toda, Toru Miyoshi, Shinsuke Yuasa

**Affiliations:** 1Department of Cardiovascular Medicine, Graduate School of Medicine, Dentistry and Pharmaceutical Sciences, Okayama University, Okayama 700-8558, Japan; p8w24uzd@okayama-u.ac.jp (K.I.); akagi-s@cc.okayama-u.ac.jp (S.A.); takayayoichi@okayama-u.ac.jp (Y.T.); hiromail1979@s.okayama-u.ac.jp (H.T.); miyoshit@cc.okayama-u.ac.jp (T.M.); yuasa@okayama-u.ac.jp (S.Y.); 2Center for Advanced Heart Failure, Okayama University Hospital, Okayama 700-8558, Japan

**Keywords:** heart failure, insulin resistance, HOMA-IR, diabetes mellitus

## Abstract

Diabetes mellitus (DM) is a major risk and prognostic factor for heart failure (HF). Insulin resistance (IR) is an important component of DM, but the relationship between IR and HF prognosis has not yet been established across a wide variety of HF populations. We retrospectively evaluated the relationship between IR and clinical outcomes of HF patients at our hospital between 2017 and 2021. IR was defined as a homeostatic model assessment of IR (HOMA-IR) index ≥ 2.5, calculated from fasting blood glucose and insulin concentrations. The primary outcome was a composite of all-cause death and hospitalisation for HF (HHF). Among 682 patients included in the analyses, 337 (49.4%) had IR. The median age was 70 [interquartile range (IQR): 59–77] years old, and 66% of the patients were men. Among the patients, 41% had a left ventricular ejection fraction below 40%, and 32% had DM. The median follow-up period was 16.5 [IQR: 4.4–37.3] months. IR was independently associated with the primary outcome (HR: 1.91, 95% CI: 1.39–2.62, *p* < 0.0001), death (hazard ratio [HR]: 1.86, 95% confidence interval [CI]: 1.28–2.83, *p* < 0.01), and HHF (HR: 1.91, 95% CI: 1.28–2.83, *p* < 0.01). HOMA-IR is an independent prognostic factor of HF in a wide variety of HF populations.

## 1. Introduction

Heart failure (HF) imposes a substantial burden on the morbidity and mortality of patients, and a growing number of HF cases threaten the healthcare system [1]. HF and diabetes mellitus (DM) often occur concomitantly, and each disease independently increases the risk of the other. In HF patients, including HF with reduced ejection fraction (HFrEF), mildly reduced ejection fraction (HFmrEF), and preserved ejection fraction (HFpEF), the DM prevalence ranges from 10% to 47% [2]. Observational studies have consistently demonstrated an increased risk of incident HF in DM patients without a history of cardiovascular disease [3]. Additionally, DM carries an adverse risk of death and hospitalisation for HF (HHF) among HF patients, including both HFrEF and HFpEF [4].

The pathological condition of insulin resistance (IR) is defined as the inability of insulin to adequately lower blood glucose concentrations, representing a leading pathogenic mechanism of DM [5]. IR is associated with a risk of developing HF after accounting for traditional risk factors, including DM [6]. In the meta-analysis, the highest tertile of homeostatic model assessment of insulin resistance (HOMA-IR), the insulin resistance index, was associated with a 45% increase in developing HF compared with the lowest tertile. Moreover, recent studies have revealed that IR was prevalent among HF patients, even in the absence of overt DM [7].

A previous study demonstrated an independent association between IR and mortality among non-DM patients with chronic HFrEF [8]. Low insulin sensitivity, which was assessed by glucose and insulin dynamic profiles during an intravenous glucose tolerance test, was associated with a 163% increase in all-cause death among chronic heart failure patients. However, the relationship between IR and prognosis among a wide variety of HF patients remains to be investigated, particularly in HFpEF, non-DM, and acute HF patients. Therefore, this study aimed to evaluate the association between IR and clinical outcomes in a wide variety of HF patients.

## 2. Materials and Methods

### 2.1. Participants and Study Design

The current study is a single-centre, retrospective, observational study on HF (UMIN000048788 at the University Hospital Medical Information Network Clinical Trials Registry). A retrospective medical record review was conducted for patients hospitalised for HF between January 2017 and December 2021 at our hospital. The diagnosis of HF was made by a physician. Patients aged <18 years were excluded. Among the patients, those with available fasting blood glucose and insulin measurements during the hospitalisation were enrolled in the analysis. The study conformed to the principles outlined in the Declaration of Helsinki, and the Ethics Committee of the Okayama University Graduate School of Medicine, Dentistry and Pharmaceutical Sciences and Okayama University Hospital approved this study (Approval Number 2209-028, Approval Date: 26 August 2022); a waiver of informed consent was obtained owing to the retrospective nature of the study.

### 2.2. Insulin Resistance Assessment

The fasting blood glucose and insulin measurements during the hospitalisation were used for the analysis. If there were multiple measurements, the first measurement of fasting blood glucose and insulin was used for the analysis. The HOMA-IR index was calculated from the fasting glucose and insulin levels using the following equation: (fasting glucose [mg/dL] × fasting insulin [uIU/mL)/405 [9]. IR was defined as HOMA-IR ≥ 2.5, based on a previous report [10].

### 2.3. Clinical Covariates and Outcomes

All the covariates were collected from medical records. Diabetes mellitus (DM) was defined as fasting plasma glucose ≥ 126 mg/dL on two occasions or fasting plasma glucose ≥ 126 mg/dL and haemoglobin A1c ≥ 6.5% on one occasion, glucose-lowering treatment, or a previous DM diagnosis. Hypertension was defined as blood pressure > 140/90 mmHg measured on several occasions or the administration of antihypertensive treatment. Dyslipidaemia was defined as the administration of lipid-lowering treatment. Renal function was assessed by calculating the estimated glomerular filtration rate (eGFR) using the modified isotope dilution mass spectrometry-traceable 4-variable Modification of Diet in Renal Disease study equation, which was modified for the Japanese population: eGFR = 194 × (serum creatinine [mg/dL]) − 1.094 × (age [years]) − 0.287 × 0.739 (if female). This equation was previously validated in the Japanese population [11]. All the patients’ nutrition condition was evaluated according to the Geriatric Nutritional Risk Index (GNRI) formula: GNRI = [1.489 × albumin (g/L)] + [41.7 × (weight/IBW)], where IBW means ideal body weight and was calculated from the following equations: 22 × H × H (H: height). We used two types of admission categories (planned and unplanned). Ischemic aetiology was defined as having a history of myocardial infarction or based on objective evidence of coronary artery disease, which is considered the main cause of HF. Chronic kidney disease was defined as eGFR < 60 mL/min/1.73 m^2^. For the clinical outcomes, all-cause death and HHF were evaluated. The primary outcome was a composite of all-cause mortality and hospitalisation for heart failure. The secondary outcomes were all-cause mortality and hospitalisation for heart failure. Clinical outcomes and follow-up periods were evaluated from the day of HOMA-IR measurement. The data on mortality and HHF were retrieved from the medical records of our hospital.

### 2.4. Statistical Analysis

All the statistical analyses were conducted using R version 4.1.2. Based on the traditional IR definition (HOMA-IR ≥ 2.5), we used two HOMA-IR categories (IR: HOMA-IR ≥ 2.5; non-IR: HOMA-IR < 2.5). Continuous data are presented as medians [quartile 1, quartile 3], and categorical data are presented as numbers and percentages. The distribution of each continuous variable was assessed using histograms. B-type natriuretic peptide (BNP) had a skewed distribution; therefore, BNP values were analysed after logarithmic transformation. Continuous and categorical variables were compared using the Mann–Whitney U and Chi-squared tests, respectively. Follow-up time was calculated using the Kaplan–Meier estimate of potential follow-up, and log-rank tests were used to compare clinical outcomes according to the HOMA-IR categories. Multivariate Cox regression proportional hazards models were used to assess the relationship between HOMA-IR categories and clinical outcomes, and we adjusted for age, sex, DM, weight, ischaemic aetiology, eGFR, log-transformed BNP level, serum sodium, sodium-glucose co-transporter 2 inhibitor (SGLT2 inhibitor), GNRI, and admission type. The covariates were selected considering a limited number of events and clinical associations to the outcomes and HOMA-IR. The following subgroups were analysed for the primary outcome: age (≤median, >median), sex, weight (≤median, >median), ischaemic or non-ischaemic aetiology, left ventricular ejection fraction (LVEF) category, DM, hypertension, dyslipidaemia, chronic kidney disease (eGFR < 60 mL/min/1.73 m2, eGFR ≥ 60 mL/min/1.73 m2), BNP (≤median, >median), GNRI (≤median, >median), and admission type. The LVEF was defined as follows: heart failure with reduced ejection fraction (HFrEF), LVEF < 40%; heart failure with mildly reduced ejection fraction (HFmrEF), LVEF 40–50%; and heart failure with preserved ejection fraction (HFpEF), LVEF ≥ 50%. Receiver operating characteristic (ROC) curve analysis was used to assess the prognostic value of HOMA-IR for the primary outcome. The proportionality assumptions of the Cox regression models were evaluated by log–log survival curves. Multiple imputations were used to manage missing BNP data. Statistical significance was set at *p* < 0.05.

## 3. Results

### 3.1. Baseline Characteristics of the Study Population

Among 996 patients hospitalised at our institution for HF, 682 patients with HOMA-IR measurements were enrolled in the analysis. Table 1 summarises the baseline patient characteristics.

The median age was 70 [interquartile range (IQR): 59–77] years, and 450 (66.0%) patients were men. Of these, 134 (19.6%) had an ischaemic aetiology. Among 282 HFrEF patients, the aetiologies of HF were as follows: 144 (51.1%) with dilated cardiomyopathy, 79 (28.0%) with ischaemic heart disease, 9 (3.2%) with valvular heart disease, 14 (5.0%) with cardiac sarcoidosis, 8 (2.8%) with the dilated phase of hypertrophic cardiomyopathy, 7 (2.5%) with adult congenital heart disease, 4 (1.4%) with cancer therapy-related cardiac dysfunction, and 8 (2.8%) with other conditions. The patients were stratified according to HOMA-IR. A total of 337 (49.4%) patients had HOMA-IR ≥ 2.5 and were categorised as IR. Table 1 presents the baseline demographic and clinical characteristics of the patients stratified into IR and non-IR groups. The patients with IR had a higher prevalence of ischaemic heart disease, DM, and HFrEF. They also had higher age, height, weight, BMI, body surface area, BNP level, GNRI, and C-reactive protein level. A higher proportion of patients with IR received beta-blockers and insulin compared to non-IR patients. There was no significant difference in the rates of unplanned hospitalisation between the two groups.

### 3.2. Clinical Outcomes According to IR Grouping

The median follow-up period was 16.5 [IQR: 4.4–37.3] months. The overall composite outcome rate during this period was 27.1% (185/682), and the survival rate by Kaplan–Meier analysis demonstrated that IR was associated with higher rates of composite of death or HHF (log-rank: *p* < 0.001; Figure 1). Multivariate Cox regression analysis after adjustment for clinical covariates demonstrated that IR was a significant predictor of an increased composite of death or HHF (hazard ratio [HR]: 1.91, 95% CI: 1.39–2.62, *p* < 0.0001, Table 2).

All-cause mortality and HHF rates were 15.8% (108/682) and 17.4% (119/682), respectively. The survival rate by Kaplan–Meier analysis demonstrated that IR was associated with higher rates of mortality and HHF (log-rank: *p* = 0.003, and *p* = 0.002, respectively; Figure 1).

Multivariate Cox regression analysis after adjustment for clinical covariates demonstrated that IR was a significant predictor of increased mortality and HHF (HR: 1.86, 95% confidence interval [CI]: 1.22–2.83; HR: 1.91, 95% CI: 1.28–2.83, respectively, Table 2). 

The ROC curve analysis revealed that the area under the curve was 0.60 (95% CI: 0.54–0.66, Figure 2).

Table 3 shows the univariate and multivariate Cox regression analyses of the common prognostic variables for HF included in the multivariate model.

In our dataset, GNRI, BNP level, and eGFR were HF predictors in both the univariate and multivariate models. Age, male sex, DM, SGLT2 inhibitor, sodium, and ischaemic heart disease were HF predictors in the univariate but not the multivariate model.

### 3.3. Association of IR with Clinical Outcomes in Each Subgroup

The prognostic impact of IR on the primary outcome across different subgroups is shown in Figure 3.

Associations between IR and the composite outcome were consistent across most subgroups, except in the subgroups defined by age, DM, and BNP level (interaction *p* values: 0.01, <0.01, and 0.03, respectively). The association between IR and the composite outcome was prominent in patients who were younger, had no DM, and had lower BNP levels (HR: 2.13, 95% CI: 1.32–3.42; HR: 2.24, 95% CI: 1.50–3.35; HR: 3.01, 95% CI: 1.68–5.39, respectively), but not those with higher age, DM, and higher BNP levels (HR: 1.46, 95% CI: 0.99–2.16; HR: 0.93, 95% CI: 0.59–1.46; HR: 1.40, 95% CI: 0.98–2.00, respectively).

## 4. Discussion

The current study showed that (1) the IR index of HOMA-IR was independently associated with death (HR: 1.86, 95% CI: 1.22–2.83, *p* < 0.01), HHF (HR: 1.91, 95% CI: 1.28–2.83, *p* < 0.01) and the composite of death or HHF (HR: 1.91, 95% CI: 1.39–2.62, *p* < 0.0001), and (2) HOMA-IR was strongly associated with clinical outcomes in patients who were younger and had no DM and lower BNP level, with significant interactions in age, DM, and BNP (*p* = 0.01, *p* < 0.01, *p* = 0.03, respectively). In this large cohort of HF patients with HOMA-IR assessment, we demonstrated that IR had a predictive value for mortality and HHF among HF patients, after accounting for traditional risk factors, including DM.

Studies evaluating the relationship between IR and HF prognosis are limited. Wolfram et al. demonstrated that the insulin sensitivity index, which was evaluated and calculated from intravenous glucose tolerance testing [12], was independently associated with mortality among non-DM patients with chronic HFrEF [8]. The prognostic value of IR among patients with HFpEF, non-DM patients, and those with chronic HF has not been adequately elucidated. Additionally, since a previous study assessed mortality alone as a clinical outcome, the impact of IR on HHF is unknown. Our HF cohort included a wide variety of HF patients, including HFrEF and HFpEF patients, patients with and without DM, and those with acute and chronic HF. HOMA-IR is easily calculated from a fasting blood sample. While this IR index is widely established as a clinical predictor of cardiovascular diseases, its predictive value has not been evaluated in HF. Using a large cohort of HF patients with HOMA-IR assessment, we demonstrated an independent predictive value of HOMA-IR for mortality and HHF among a wide variety of HF patients. The HR in this study was comparable to that of a previous study on non-DM patients with chronic HF (reported as 2.63 in a multivariate model) [8]. Compared to the previous study, our study cohort had older age and includes DM patients. Significant interactions of the effect of age and DM to the relationship between IR and clinical outcomes may explain the difference.

The independent association between IR and mortality has been explored in various populations, including patients with type 1 DM [13], type 2 DM [14], and non-DM patients [15,16,17,18]. However, discrepancies exist among these studies regarding the relationship between IR and mortality, with either the presence or absence of a significant association between IR and all-cause mortality. Our results demonstrated a significant association between IR and all-cause mortality in the HF population, consistent with previous results in non-DM patients with chronic HFrEF [8]. However, this relationship was inconsistent among the subgroups. In this regard, subgroup analysis in the current study demonstrated a significant interaction in the subgroups defined by age, DM, and BNP level. Although the precise reasons for the inconsistency in these subgroups remain unknown, previous studies have reported some inconsistencies in subgroup analysis. The 14 variations in the study population, increased rates of non-cardiac death in elderly patients, and diabetic medications in DM patients might partially explain the inconsistencies in the relationship between IR and clinical outcomes.

The current study did not have sufficient data to demonstrate the precise mechanisms of IR association with subsequent mortality and HHF. However, the interdependence between HF and glucose metabolism may be partially involved in this relationship. HF and DM often occur concomitantly, and each disease independently increases the risk of the other. For example, John et al. demonstrated that DM was associated with a nearly two-fold increase in the risk of incident HF (HR: 1.74, 95% CI: 1.38–2.19), even after adjustment for other cardiovascular risk factors [19]. However, among non-DM patients with HF enrolled in the Candesartan in Heart Failure-Assessment of Reduction in Mortality and Morbidity (CHARM) Program and the Eplerenone in Mild Patients Hospitalization and Survival Study in HF (EMPHASIS-HF) trial, the incidence of DM was 28 and 21 per 1000 person-years, respectively, which was substantially higher than that in adults of similar age in the general population (9.4–10.9 per 1000 person-years for adults aged 45 years), indicating that HF predisposed the onset of DM [2]. Further, HF improvement by a left ventricular assist device reportedly corrected glycaemic control and IR in HF patients [20]. Thus, glucose metabolism and HF status were interdependent. In fact, a previous study demonstrated that IR was dependent on whether the HF was acute decompensated or chronic stable [21]. Therefore, the association between IR and clinical outcomes in the current study may reflect HF status, which was not completely adjusted by conventional HF prognostic predictors.

It is important to determine (1) whether IR has a causal relationship with clinical adverse outcomes in HF, and whether correcting IR has beneficial effects, or (2) whether IR does not have a causal relationship and has only predictive value. Several small randomised, placebo-controlled trials have been conducted to evaluate the clinical benefit of metformin in correcting IR among HF patients [22,23]. Although there was no significant improvement in peak oxygen uptake among the metformin groups, these trials showed significant improvement in the slope of the ratio of minute ventilation to carbon dioxide production and the work metabolic index. Further studies are needed to assess the clinical benefits of IR correction in HF patients.

Several important limitations of this study should be mentioned. First, the current study was a single-centre, retrospective, observational study that did not necessarily represent HF patients in other parts of the world. Therefore, our results should be considered hypothesis-generating, and further multi-centre validation studies are warranted. Second, due to the nature of observational studies, this study revealed only an association between IR and clinical outcomes but not a causal relationship between them. Third, while HOMA-IR levels were reported during the fasting state, some subjects might not have fasted as requested, and their HOMA-IR levels might not have been representative of their true glycaemic state. Finally, information regarding clinical parameters and drug therapy was based on digitised data, which must be considered.

## 5. Conclusions

In conclusion, our observational study demonstrated that IR was highly prevalent and had independent associations with subsequent all-cause mortality, HHF, and the composite of death or HHF among a wide variety of HF patients.

## Figures and Tables

**Figure 1 nutrients-16-01888-f001:**
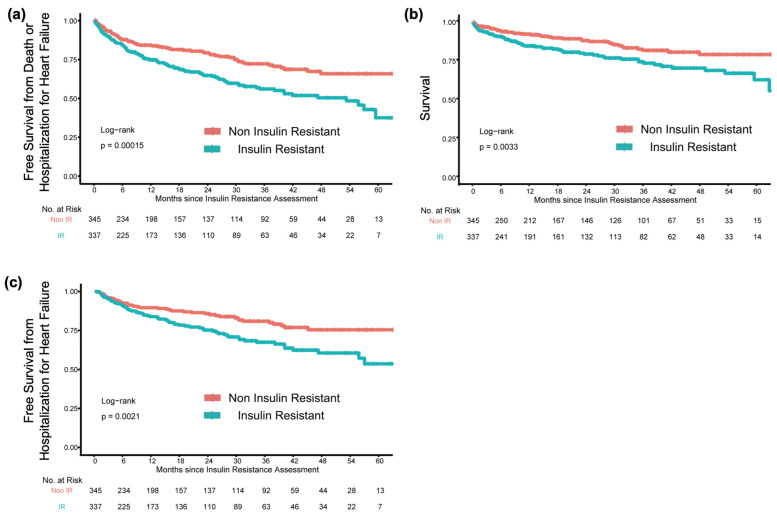
Clinical outcomes according to insulin resistance. The event-free survival rates according to IR, defined as HOMA-IR ≥ 2.5, are from the composite of mortality or hospitalisation for HF (**a**), mortality (**b**), and hospitalisation for HF (**c**).

**Figure 2 nutrients-16-01888-f002:**
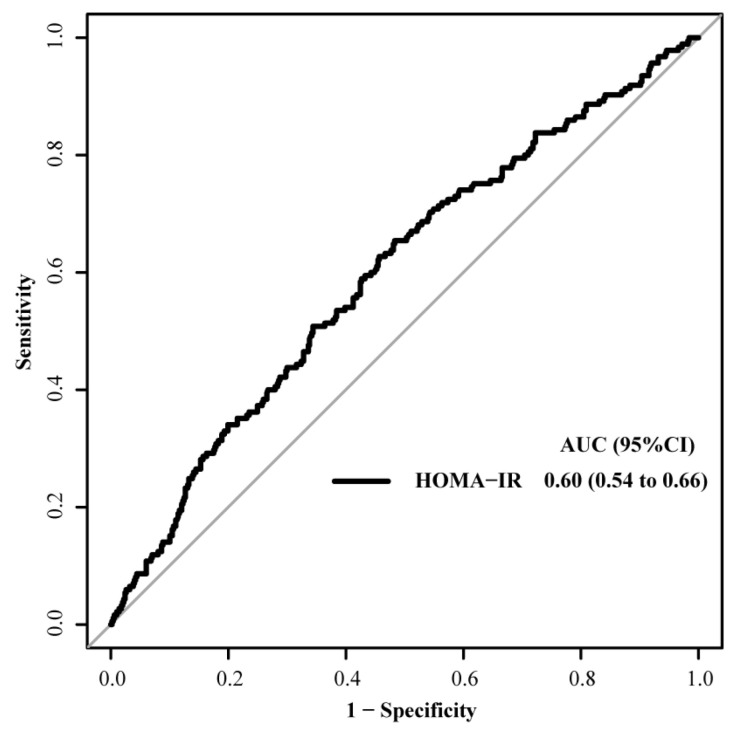
A receiver operating characteristic curve analysis of HOMA-IR for the primary outcome.

**Figure 3 nutrients-16-01888-f003:**
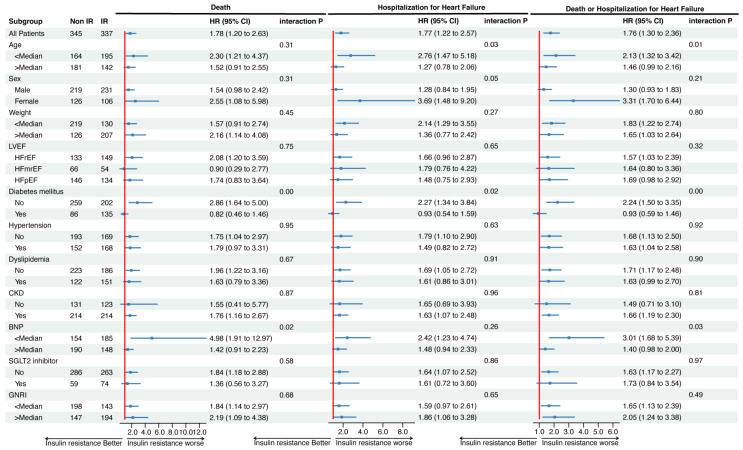
Association between IR and clinical outcomes in each subgroup. Abbreviations: CKD, chronic kidney disease; SGLT2 inhibitor, sodium-glucose co-transporter 2 inhibitor.

**Table 1 nutrients-16-01888-t001:** Baseline demographic and clinical characteristics.

	All	HOMA-IR < 2.5	HOMA-IR ≥ 2.5	*p* Value
n	682	345	337	
Age (median [IQR])	70 [59–77]	71 [61–78]	68 [57–76]	0.004
Male sex (%)	450 (66.0)	219 (63.5)	231 (68.5)	0.19
Height (median [IQR])	162 [155, 168]	161 [153, 167]	163 [156, 169]	0.005
Weight (median [IQR])	60 [51, 70]	57 [48, 65]	64 [54, 74]	<0.001
BSA (median [IQR])	1.63 [1.49–1.77]	1.58 [1.45–1.73]	1.69 [1.56–1.82]	<0.001
BMI (median [IQR])	22.8 [20.3–25.7]	21.8 [19.8–24.1]	24.3 [21.3–27.1]	<0.001
Ischaemic aetiology (%)	134 (19.6)	47 (13.6)	87 (25.8)	<0.001
Hypertension (%)	320 (46.9)	152 (44.1)	168 (49.9)	0.15
Dyslipidaemia (%)	273 (40.0)	122 (35.4)	151 (44.8)	0.02
DM (%)	221 (32.4)	86 (24.9)	135 (40.1)	<0.001
AF or AFL (%)	134 (19.6)	67 (19.4)	67 (19.9)	0.96
CKD (%)	428 (62.8)	214 (62.0)	214 (62.5)	0.75
CIED (%)	200 (29.3)	104 (30.1)	96 (28.5)	0.70
CRT (%)	95 (13.9)	46 (13.3)	49 (14.5)	0.73
Unplanned hospitalisation (%)	257 (37.7)	137 (39.7)	120 (35.6)	0.31
LVEF spectrum				0.28
HFrEF (%)	282 (41.3)	133 (38.6)	149 (44.2)	
HFmrEF (%)	120 (17.6)	66 (19.1)	54 (16.0)	
HFpEF (%)	280 (41.1)	146 (42.3)	134 (39.8)	
LVEF (median [IQR])	44 [31–58]	45 [33–59]	43 [30–57]	0.22
LVDd (median [IQR])	53 [46–60]	52 [46–59]	54 [46–61]	0.08
LAD (median [IQR])	43 [39–50]	43 [39–49]	44 [40–50]	0.12
BNP (median [IQR])	371 [148–778]	447 [201–900]	297 [117–646]	<0.001
eGFR (median [IQR])	51 [35–68]	52 [35–68]	50 [34–67]	0.50
Albumin (median [IQR])	3.9 [3.4–4.2]	3.8 [3.4–4.2]	3.9 [3.4–4.2]	0.36
Sodium (median [IQR])	139 [137–141]	139 [137–141]	139 [137–141]	1.00
CRP (median [IQR])	0.19 [0.08–0.66]	0.16 [0.06–0.59]	0.22 [0.10–0.74]	0.007
GNRI	100.8 [92.4, 109.4]	98.2 [90.6, 105.5]	103.5 [94.7, 112.3]	<0.001
HOMA-IR (median [IQR])	2.47 [1.41–5.64]	1.42 [0.93–1.84]	5.72 [3.52–10.50]	<0.001
RAAS inhibitor (%)	421 (61.7)	202 (58.6)	219 (65.0)	0.10
ACEI (%)	281 (41.2)	133 (38.6)	148 (43.9)	0.18
ARB (%)	115 (16.9)	55 (15.9)	60 (17.8)	0.59
ARNI (%)	26 (3.8)	14 (4.1)	12 (3.6)	0.89
BB (%)	554 (81.2)	267 (77.4)	287 (85.2)	0.01
MRA (%)	405 (59.4)	207 (60.0)	198 (58.8)	0.80
SGLT2 inhibitor (%)	133 (19.5)	59 (17.1)	74 (22.0)	0.13
Statin (%)	229 (33.6)	90 (26.1)	139 (41.2)	<0.001
Insulin (%)	35 (5.1)	4 (1.2)	31 (14.5)	<0.001

Abbreviations: ACEI, angiotensin-converting enzyme inhibitor; AF, atrial fibrillation; AFL, atrial flutter; ARB, angiotensin receptor blocker; ARNI, angiotensin receptor neprilysin inhibitor; BB, beta blocker; BMI, body mass index; BSA, body surface area; CIED, cardiac implantable electronic device; CKD, chronic kidney disease; CRP, C-reactive protein; CRT, cardiac resynchronisation therapy; DM, diabetes mellitus; GNRI, Geriatric Nutritional Risk Index; HFmrEF, heart failure with mildly reduced ejection fraction; HFpEF, heart failure with preserved ejection fraction; HFrEF, heart failure with reduced ejection fraction; LAD, left atrial diameter; LVDd, left ventricular diastolic diameter; MRA, mineralocorticoid receptor antagonists; RAAS inhibitor, renin angiotensin aldosterone system inhibitor; SGLT2 inhibitor, sodium-glucose co-transporter 2 inhibitor.

**Table 2 nutrients-16-01888-t002:** Univariate and multivariate Cox proportional hazard analyses of IR for clinical outcomes.

	Univariate			Multivariate		
	Hazard Ratio	95% CI	*p* Value	Hazard Ratio	95% CI	*p* Value
Death	1.78	1.20 to 2.63	0.00461	1.86	1.22 to 2.83	0.0041
HHF	1.77	1.22 to 2.57	0.00293	1.91	1.28 to 2.83	0.00162
Death or HHF	1.76	1.30 to 2.36	0.000247	1.91	1.39 to 2.62	0.0000778

The multivariate model included the following variables: HOMA-IR > 2.5, age, sex, weight, DM, ischaemic aetiology, log(BNP), sodium, SGLT2 inhibitor, eGFR, and GNRI. Abbreviations: HHF, hospitalisation for heart failure; CI, confidence interval; SGLT2 inhibitor, sodium-glucose co-transporter 2 inhibitor.

**Table 3 nutrients-16-01888-t003:** Univariate and multivariate COX proportional analyses of other covariates for the composite outcome.

	Univariate			Multivariate		
	Hazard Ratio	95% CI	*p* Value	Hazard Ratio	95% CI	*p* Value
Age	1.02	1.01 to 1.04	<0.001	1.01	1.00 to 1.02	0.10
Male sex	1.58	1.13 to 2.21	<0.01	1.36	0.90 to 2.07	0.14
Weight	0.99	0.95 to 1.02	0.41	1.00	0.98 to 1.02	0.95
DM	1.68	1.25 to 2.25	<0.001	1.10	0.78 to 1.54	0.60
Ischaemic aetiology	1.94	1.41 to 2.66	<0.001	1.17	0.82 to 1.67	0.39
SGLT2 inhibitor	1.49	1.03 to 2.13	0.03	1.30	0.88 to 1.92	0.19
GNRI	0.96	0.94 to 0.97	<0.001	0.98	0.96 to 0.99	<0.01
Sodium	0.92	0.89 to 0.96	<0.001	0.96	0.93 to 1.00	0.07
log(BNP)	1.51	1.32 to 1.72	<0.001	1.21	1.04 to 1.41	0.01
eGFR, per 1 mL/min/1.73 m^2^	0.98	0.97 to 0.98	<0.001	0.99	0.98 to 0.99	<0.001

The multivariate model includes following variables: HOMA-IR ≥ 2.5, age, sex, weight, dm, ischemic etiology, log(bnp), egfr, sodium, SGLT2 inhibitor, GNRI. Abbreviations: 95% CI = 95% confidence interval; GNRI = geriatric nutritional risk index.

## Data Availability

The research data are unavailable due to privacy and ethical restrictions.

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
