# Peer review of "Prognostic Implications of Insulin Resistance in Heart Failure in Japan"

_nutrients, 2024, doi:10.3390/nu16121888_

Round 1

Reviewer 1 Report (New Reviewer)

Comments and Suggestions for Authors

The manuscript approached the importance and prognostic implications of insulin resistance in patients with heart failure. Diabetes mellitus and, generally, insulin resistance are well-known predictors of poor outcome in cardiovascular diseases, and specifically in HF. The mutual negative influence of each pathology in the other makes this a hot topic for the heart team management.

The included cohort is consistent, the statistic analysis is solid, albeit the final results (and subsequent conclusions) are somehow expected.

-        Line 115: correct the value for LVEF (≥50%)

-        In table I, reconsider the LVEF variable (is mentioned only LVEF…is it HFrEF, HFmREF or HFpEF?) it is not clear…

-        Please provide additional details regarding the influence of SGLT2 inhibitors in the prognosis of patients with HF with preserved EF (see DELIVER study) with or without IR. The negative prognosis associated with IR may be improved by the concomitant administration of SGLT2i? Or the prognosis is per se influenced only by the HFpEF?

-        The baseline characteristics lacks details concerning the inflammatory profile (C-reactive protein) and the use of statins (as an important share of patients presented with dyslipidemia). These aspects are important as they can influence not only the metabolic profile (i.e. IR) but also the evolution and outcome of HF.

Best regards

Comments on the Quality of English Language

English is generally fine

Author Response

We greatly appreciate the reviewer’s comments.

Comment:

- Line 115: correct the value for LVEF (≥50%)

- In table I, reconsider the LVEF variable (is mentioned only LVEF…is it HFrEF, HFmREF or HFpEF?) it is not clear…

Response: Thank you for your comment. In accordance with the reviewer’s comment, we have corrected the line 115 and have added the following sentences in line 118: “The LVEF were defined as follows; heart failure with reduced ejection fraction (HFrEF), LVEF <40%; heart failure with mildly reduced ejection fraction (HFmrEF), LVEF 40-50%; heart failure with preserved ejection fraction (HFpEF), LVEF ≥50%.” (line 118 in Material and Methods) and Table 1 Explannation: “HFmrEF, heart failure with mildly reduced ejection fraction; HFpEF, heart failure with preserved ejection fraction; HFrEF, heart failure with reduced ejection fraction;” (line 135 in Table 1 Explannation).

Comment:

- Please provide additional details regarding the influence of SGLT2 inhibitors in the prognosis of patients with HF with preserved EF (see DELIVER study) with or without IR. The negative prognosis associated with IR may be improved by the concomitant administration of SGLT2i? Or the prognosis is per se influenced only by the HFpEF?

Response: Thank you for your comment. We analyzed the influence of SGLT2 inhibitors in HFpEF population as follows.

SGLT2I

There was no significant interaction of SGLT2 inhibitors to the association between IR and clinical outcomes. However, as our HFpEF cohort included only 34 patients with SGLT2 inhibitor, these results should be carefully interpreted for small sample size. Therefore, we did not comment on these results in the manuscript.

Comment:

- The baseline characteristics lacks details concerning the inflammatory profile (C-reactive protein) and the use of statins (as an important share of patients presented with dyslipidemia). These aspects are important as they can influence not only the metabolic profile (i.e. IR) but also the evolution and outcome of HF.

Response: Thank you for your comment. We have added CRP and the use of statins in table1.

Reviewer 2 Report (New Reviewer)

Comments and Suggestions for Authors

This study by Iwasaki K et al. was designed to investigate the relationship between insulin resistance (IR) and heart failure (HF) prognosis. Total 996 patients were hospitalised for HF at their institution in Okayama, and 682 patients with HOMA-IR measurements were enrolled in this analysis. The authors concluded that (1) the IR index for HOMA-IR was strongly associated with death, hospitalization for HF (HHF), and a composite of death or HHF; and (2) HOMA-IR was strongly associated with clinical outcomes in younger patients, had no DM, and had low levels of BNP, with a significant interaction between age, DM, and BNP. In this manuscript, the introduction presents the context of the research model and articulates the research questions and objectives. The results section is clear, well-organized, and descriptively robust. The discussion is correct and appropriately compared with the literature. Limitations are properly described. Overall, this report is clear, comprehensive, and well-organized.

Minor recommendation,

(1)    Since this is a single-center observational study, indicating the country in the title would be more appropriate.

(2)    There is a discrepancy between lines 14 (between 2016 and 2020) and 58 (between January 2017 and December 2021), please clarify when patients were enrolled in this study.

(3)    Typo problem in line 160, “he” event-free survival rates…

Author Response

Reviewer 2

We greatly appreciate the reviewer’s comments.

Comment:

  • Since this is a single-center observational study, indicating the country in the title would be more appropriate.

Response: Thank you for your comment. In accordance with the reviewer’s comment, we have added the country name in the title: “Prognostic Implications of Insulin Resistance in Heart Failure in Japan”.

Comment:

  • There is a discrepancy between lines 14 (between 2016 and 2020) and 58 (between January 2017 and December 2021), please clarify when patients were enrolled in this study.

Response: Thank you for the comment. We apologize for the inappropriate sentence. We have corrected the text in line 14 of the previous draft to read "between 2017 and 2021" (line 15 of the new draft).

Comment:

  • Typo problem in line 160, “he” event-free survival rates…

Response: Thank you for the comment. We have corrected the word in line 160 of the previous draft (line 165 of the new draft).

This manuscript is a resubmission of an earlier submission. The following is a list of the peer review reports and author responses from that submission.

Round 1

Reviewer 1 Report

Comments and Suggestions for Authors

This is a retrospective study that examined the relationship between insulin resistance (IR) and heart failure prognosis across a wide variety of heart failure populations. The primary outcome was a composite of all-cause death and hospitalisation for heart failure (HHF). Among 682 patients included in the analyses, 49.4% had IR, 41% had a %EF below 40%, and 32% had diabetes. The median follow-up period was 16.5 months. IR was independently associated with the primary outcome, death and HHF. It is concluded that HOMA-IR is independent prognostic factors of heart failure in a wide variety of heart failure populations.

General Comments:

Previous studies have shown that insulin resistance (IR) is associated with heart failure severity.  Therefore, this study is not that novel.  However, this study does show that hospitalization for heart failure (HHF) is the main adverse event associated with IR.

Not unexpected, the incidence of diabetes mellitus (DM) was substantially higher in the IR patients.  A similar analysis should also be done examining the effects of DM on death and HHF to determine if it is IR or DM that is the major predictor of poor heart failure outcomes.

Specific Comments:

1)    A weakness of this study is that it is retrospective and was performed at a single center.  However, the authors do recognize this.

2)    When was the HOMA-IR measured?  Were serial measurements made?  Were there any changes in HOMA-IR over the 60-month study period?

3)    Table 1:  What does the LVEF spectrum mean?  It is all HFrEF (%).  I believe it should be HFrEF, HFmrEF and HFpEF.

4)    Table 1: Abbreviation for DM (probably diabetes mellitus) should be defined in Table legend.

Figure 3: It is mainly HHF driving correlation between IR and outcomes.  It should be highlighted that IR may not be that good at predicting death from heart failure.